# Endometriosis: Update of Pathophysiology, (Epi) Genetic and Environmental Involvement

**DOI:** 10.3390/biomedicines11030978

**Published:** 2023-03-22

**Authors:** Nicolas Monnin, Anne Julie Fattet, Isabelle Koscinski

**Affiliations:** 1Majorelle Clinic, Atoutbio Laboratory, Laboratory of Biology of Reproduction, 54000 Nancy, France; 2Laboratory of Biology of Reproduction, Hospital Saint Joseph, 13008 Marseille, France; 3NGERE Inserm 1256, 54505 Vandoeuvre les Nancy, France

**Keywords:** endometriosis, epigenetics, molecular pathophysiology, endocrine disruptor

## Abstract

Endometriosis is a chronic disease caused by ectopic endometrial tissue. Endometriotic implants induce inflammation, leading to chronic pain and impaired fertility. Characterized by their dependence on estradiol (via estrogen receptor β (ESRβ)) and their resistance to progesterone, endometriotic implants produce their own source of estradiol through active aromatase. Steroidogenic factor-1 (SF1) is a key transcription factor that promotes aromatase synthesis. The expression of *SF1* and *ESRβ* is enhanced by the demethylation of their promoter in progenitor cells of the female reproductive system. High local concentrations of estrogen are involved in the chronic inflammatory environment favoring the implantation and development of endometriotic implants. Similar local conditions can promote, directly and indirectly, the appearance and development of genital cancer. Recently, certain components of the microbiota have been identified as potentially promoting a high level of estrogen in the blood. Many environmental factors are also suspected of increasing the estrogen concentration, especially prenatal exposure to estrogen-like endocrine disruptors such as DES and bisphenol A. Phthalates are also suspected of promoting endometriosis but throughmeans other than binding to estradiol receptors. The impact of dioxin or tobacco seems to be more controversial.

## 1. Introduction

Endometriosis is a common and benign gynecological disease defined by the ectopic presence of tissue with the same morphological and functional characteristics as the endometrium (cylindrical glandular epithelium and stroma) [1,2,3,4]. Its main locations are the pelvic peritoneum, uterosacral ligaments, cul-de-sac of Douglas, rectovaginal septum, and ovaries [1,3,4,5]. Clinical signs are numerous. Among them, dysmenorrhea, dyspareunia, chronic pelvic pain, irregular uterine bleeding and/or infertility are frequently found [3,4,5]. This ectopic endometrium is functional and undergoes periodic revisions, explaining the cyclical nature of symptoms and the chronic inflammatory process associated with estrogens [6]. This gynecological disease affects approximately 5–15% of women of reproductive age and is found in 35–50% of women with infertility [1,2,7,8,9]. Consequently, it is a frequent reason for subfertile couples to seek consultation in assisted reproductive technology (ART) centers. The pathophysiology of endometriosis is based on its “estrogen-dependent” character; however, the mechanisms of onset and development are unclear. An environmental cause is not excluded, especially via endocrine-disrupting substances that are abundant in our 21st century environment.

This review points out that local hyperestradiolaemia is the cause and consequence of endometriotic lesions’ development from epigenetically modified cells. It explains the possible modulation of hyperestradiolaemia through environmental factors, including the microbiota, how hyperestradiolaemia is involved in the distortion of the immune system, and how it can promote cancer.

## 2. Theories on Endometriosis

Several theories lead to two major pathophysiological hypotheses. The first is an endometrial origin of endometriosis implants. The other is based on an ectopic origin. In addition, risk factors and genetic predisposition factors are being studied [5].

The long-held favored group of theories rests on an endometrial origin of the endometriosis process. The benign metaplasia theory consists of the spread of endometrial cells in lymphatic and hematogenous ways. Different microvascular studies support this concept as well as the existence of some rare extra pelvic localizations, such as the bone, brain or lung, that have been histologically proven [4,5].

The coelomic metaplasia theory, embryonic Müllerian cell theory and bone marrow stem cell theory are part of the group of ectopic theories. These various possible explanations of the origin of endometriosis are based on a transformation of a tissue other than the endometrial tissue into endometriotic tissue under the action of some unknown substances [2,4,5,10,11].

Demonstrated for the first time in 1920 by Sampson, retrograde menstruation theory, in which endometrial cells are drained into the fallopian tube and peritoneal cavity during menstruation, has largely been studied. In particular, in the 1980s, Halme et al. highlighted the existence of menstrual blood in the peritoneal fluid of more than 90% of healthy women [2,4,5,7,8,12,13]. In addition, explorations carried out in women with congenital obstruction or animal experiments mimicking iatrogenic obstruction showed that obstruction of the normal menstruation flow promotes the development of endometriotic lesions. The prevalence of endometriosis also appears to be greater in women with cervical stenosis [4,5]. The organic localization of endometriotic lesions is an additional argument in favor of this theory. Indeed, superficial implants are more often located in the posterior pelvic compartment and at the left hemi-pelvis. The predisposition of lesions localized in the Douglas cul-de-sac would be explained by the accumulation of retrograde menstruation in this location under the action of gravity. Moreover, a retroverted uterus (permitting flow from the front to the rear in the vertical or lying position) is correlated with the development of endometriosis. Similarly, acting as a barrier to diffusion of menstrual flow from the left fallopian tube, a prominent sigmoid colon promotes stasis of this flow, extending the range of implantation of refluxed endometrial fragments in the left hemipelvis [5]. Finally, in a mouse model, the activation of the K-ras oncogene in endometrial cells deposited on the peritoneum allowed the development of peritoneal lesions of endometriosis, whereas the activation of this oncogene directly in peritoneal cells had no impact. This finding confirmed the endometrial origin of endometriosis [5]. Nevertheless, this theory cannot explain the complete development of endometriotic lesions by itself. Other factors, such as immune escape, adhesion to peritoneal epithelium and invasion, the neurovascular environment and growth/continuing survival, are essential to the long-term persistence of lesions. The need for these additional factors explains why only 10% of women with retrograde menstruation (90% of the general population) suffer from endometriosis [4,5,8,14].

### 2.1. An Anatomical Predisposition to Endometriosis?

The body’s inability to remove endometriosis implants into the peritoneal fluid may be aggravated by anatomical features often found in women with endometriosis. In addition to lean size [15], some elements increase menstrual reflux, hypertension of the utero-tubal junction, waves of retrograde tubular contractions of the myometrium, and uterine malformations. Moreover, in patients suffering from endometriosis, menstruation is often longer and more abundant, and menstrual cycles are shorter [2].

### 2.2. Molecular Pathophysiological Mechanisms

Similar to tumor cells, endometriotic cells have a high survival potential and clonal dissemination.

The abnormally high survival potential of endometrial cells has been the subject of several recent studies. On the one hand, genetic alterations (polygenic) have been identified as the cause of greater survival of endometrial cells present in endometriotic lesions. In particular, overexpression of the antiapoptotic *BCL-2* gene was found, promoting the proliferation of endometrial cells. These cells are also exposed to DNA damage due to their rapid turnover and their sensitivity to different epigenetic factors as well as oxidative stress. On the other hand, the high resistance of endometrial cells to apoptosis has been explored. The specific expression of the heat shock proteins (HSPs) has been highlighted. These proteins normally play a role in the protection of the correct three-dimensional folding of proteins despite thermal shocks. Endometriosis cells present a special pattern of HSPs with high expression of HSP27 and HSP70, contributing to their protection against apoptosis [2].

A recent transcriptional approach [16] comparing endometriotic cells to healthy pelvic and ovarian cells highlighted transformations at the cellular scale resulting from the reprogramming of endometriotic cells. In particular, these cells present deregulation of the pro-inflammatory pathways and over-expression of complement factors. Moreover, neo-mutations in endometriotic epithelial cells of two cancer-driver genes, *ARID1A* and *KRAS*, would favor their diffusion. KRAS is a small GTPase known to increase the proliferation potential of cells. The deciphering of disturbed pathways suggests that *ARID1A* mutation in endometriotic cells promotes the growth of local lymphatic endothelial cells through paracrine secretion of vascular growth factors. Interestingly, ovarian tumor cells associated with endometriosis present similar mutations in *KRAS*, *ARID1A*, and *PIK3CA* (a subunit of a kinase involved in tumorigenesis) [17]. PIK3CA is enrolled in the first step of activation of the PI3K/Akt pathway, which increases cell survival by blocking the apoptosis pathway, enhances cell proliferation via cyclin activation, upregulates glucose metabolism, and regulates negatively anabolic processes via inhibition of the mTOR pathway. PI3K also interacts with the PTEN pathway controlling DNA repair, genomic stability, and apoptosis [18]. Therefore, a mutation in *PIK3CA* may promote tumorigenesis.

Mutations in *KRAS*, *ARID1A*, *PIK3CA*, and others allow the studying of the clonality of various types of endometriotic lesions [19]. The redundancy in mutations within the same gene and lesions comforts the oligo-clonal character of the disease: multiple epithelial clones migrate together, especially in deep infiltrating lesions, whereas ovarian endometriomas present the highest potential for oligoclonality. These data suggest that ovarian stroma provides perfect conditions for the proliferation of multiple clones, with an increased risk of malignancy [19].

Genetic analysis of endometriotic lesions could contribute to a better diagnosis and, according to mutation profiles, could open new ways of personalized care. For instance, drugs targeting PIK3CA or MEK signals, such as alpelisib or trametinib, respectively, may theoretically offer a new treatment for women with endometriosis whose lesions have mutations in *PIK3CA* or *KRAS* [20].

### 2.3. The Crucial Role of an Altered Hormonal Environment

In patients with endometriosis, inflammation, the immune response, angiogenesis and apoptosis are altered. These disturbances are mainly caused by changes in the estrogen/progesterone environmental balance. In particular, future endometriotic cells present an increased production of estrogen and prostaglandins and develop progesterone resistance [4].

The entire pathophysiological molecular mechanism is summarized in Figure 1.

Endometriosis is an estrogen-dependent disease that is known to be “estradiol-dependent”. Estrogens act on endometrial cells via the estradiol (E2) receptors ESRα (or ESR1) and ESRβ (or ESR2). Bulun et al. have shown increased expression of ESRβ in endometriosis tissue due to the hypomethylation of the promoter region of *ESRβ*. This receptor is active through the RAS-like estrogen-regulated growth inhibitor (RERG), which induces the regulation of a large number of factors involved in resistance to apoptosis and cell proliferation [4,5,7,14,21,22].

**Figure 1 biomedicines-11-00978-f001:**
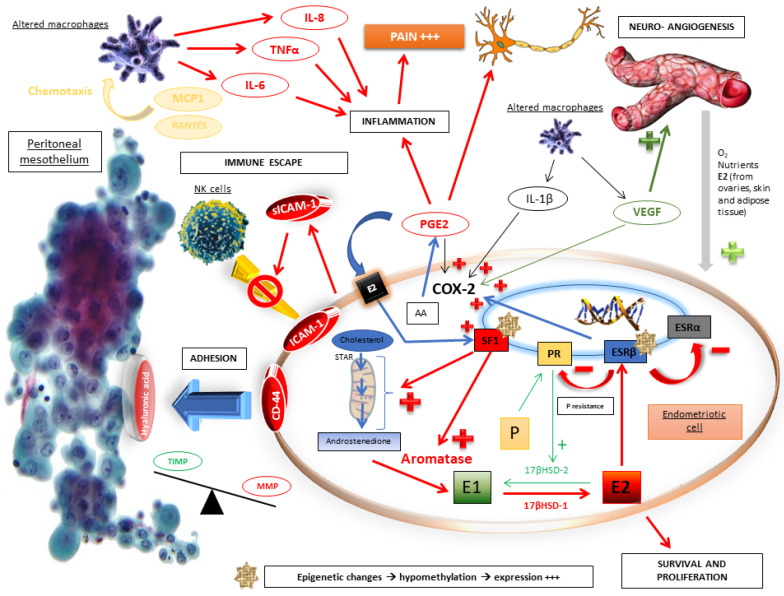
Molecular pathophysiological mechanisms of endometriosis [2,4,5,23].

In normal endometrial tissue, the production of estrogen from C19 steroids is very low due to the absence of the enzyme aromatase. Similarly, the enzyme 17β-hydroxysteroid dehydrogenase 2 (17βHSD2) (progesterone-dependent) catalyzes, during the luteal phase, the conversion of biologically highly active E2 into less active estrone (E1). Therefore, in a healthy endometrium, estrogen activity is maintained at a low level. In contrast, in the case of endometriosis, aromatase activity is detected at a high level in the endometrium as well as in ectopic endometriotic tissue. Furthermore, cells in endometriotic lesions express all the genes of steroidogenesis, including aromatase, and produce their own source of E2 from cholesterol. In addition, this high E2 concentration in these tissues is maintained due to a decrease in its catabolism as a result of deficient 17βHSD2 activity [2,4,7].

In women with endometriosis, the high level of E2 promoting endometriosis is also due to exogenous contribution, as evidenced by the high secretion of estrogen found in the ovaries, skin and adipose tissue. The E2 secreted by the ovaries reaches the endometriotic tissue through the blood. This phenomenon is mostly observed during the ovulatory phase (follicular rupture causes the release of large amounts of E2). In adipose tissue and skin, the presence of aromatase allows the conversion of circulating androstenedione into E1, which can be converted into E2 in these same tissues. The secreted E2 reaches the endometriosis implants through the blood [4].

Furthermore, in endometriotic tissue, inflammation and the production of estrogen are interconnected in an amplification loop: the oxidative stress associated with the inflammation process promotes, via an epigenetic mechanism, the overexpression of key genes of steroidogenesis (notably aromatase) and cyclooxygenase 2 (COX-2), which results in the local and continuous production of E2 and prostaglandin E2 (PGE2), respectively [4,7]. In return, prostaglandins, which are locally produced, are responsible for inflammation and pain.

While the activity of COX-2 and the production of PGE2 are low in the healthy endometrial tissue, in the endometrium of women with endometriosis and endometriotic implants, PGE2 and PGF2α are produced in excess (Figure 2). The vasoconstrictor properties of PGF2α are a cause of dysmenorrhea, and PGE2 can directly induce painful nerve stimulation, causing chronic pelvic pain. These high levels of PGE2 and PGE2α result from the high activity of prostaglandin F synthase and microsomal prostaglandin E synthase in uterine cells, which catalyze the conversion of PGH2 into PGF2α and PGE2, respectively. These enzymes, as well as COX-2, are more active in women with endometriosis than in healthy women. Four cytokines/hormones allow these higher levels of activity in endometriotic stromal cells: IL-1β cytokine, PGE2 itself (autocrine), VEGF and E2 (via ESRβ) [4,7].

By binding to the specific membrane receptors (EP1, EP2, EP3 and EP4) of endometriotic cells, PGE2 stimulates the expression of all steroidogenesis genes necessary for E2 synthesis from cholesterol. More specifically, the binding of PGE2 to its membrane receptor causes an increase in intracellular cAMP levels in endometriotic cells. This phenomenon promotes the action of a key transcription factor, the nuclear receptor of steroidogenic factor-1 (SF1), present only in endometriotic cells. SF1 enhances the expression of *STAR*(facilitating entry of cholesterol into mitochondria) and *CYP19A1*(coding for aromatase), leading to an increase in aromatase activity and, finally, a hyperestrogenic environment. In contrast, at a high rate in healthy endometrial cells, transcription inhibitors of the *STAR* and *CYP19A1* genes constitute a safety system for limiting the expression of these steroidogenesis enzymes. Among these inhibitors are the chicken ovalbumin upstream promoter-transcription factor (COUP-TF), Wilms’ tumor transcription factor 1 (WT1) and CCAAT/enhancer binding protein β (C/EBPβ). These inhibitors would be present in lower abundances in endometriosis cells [4,7,23,24].

Furthermore, the extension of endometriotic lesions is promoted by an increased level of oxytocin, a hypothalamic hormone inducing the production and release of PGE2 and PGF2a by endometrial cells and uterine hyperperistalsis. Oxytocin activates the inflammatory immune system of the endometrium and participates in an enhanced suction of debris and infectious particles from the uterine cavity as well as endometriotic implants toward the peritoneal cavity [2].

Progesterone and its various receptor isoforms (PR-A and PR-B) also play a key role in the pathophysiology of endometriosis. Physiologically, progesterone induces the differentiation of stromal endometrial cells and epithelial cells, resulting in an increased production of glycodelin (an epithelial glycoprotein produced by the secretory endometrium in the luteal phase). Glycodelin exerts indirect antiestrogenic effects. Its binding to specific PRs stimulates the synthesis of retinoic acid and increases the expression of 17βHSD2, leading to an increased conversion of E2 into the less active E1 [4].

In women with endometriosis, the response to progesterone is clearly reduced in endometrial cells because of reduced expression of epithelial glycodelin and decreased levels of progesterone. In addition, genes coding for PR are expressed in the early phase of the menstrual cycle (suggesting a progesterone resistance phenotype), and the expression of progesterone-dependent genes in the luteal phase is dysregulated [5]. PR isoform-A, an inhibitory isoform of PR, is the unique isoform expressed at high levels by endometriotic cells, regardless of cycle phase [14]. Progesterone resistance could therefore be explained by the excessive presence of the PR-A isoform and by the absence of the PR-B isoform (the active form of PR) [5,7].

Furthermore, the high expression of this inhibitory PR associated with the lack of active PR results in progesterone resistance and, ultimately, in reduced 17βHSD2 activity. This leads to a decreased conversion of E2 to E1 and, finally, the hyperestrogenic activity found in women with endometriosis (Figure 3) [4].

However, this progesterone resistance is related to the ambiguity of the role of progesterone in endometriosis pathophysiology: despite having fewer PR receptors than healthy endometrium, endometriotic tissue has a great capacity for progesterone production, which induces the physiological differentiation of endometrial stromal cells [4].

As mentioned above, nuclear SF1 receptor, only present in endometriotic cells, is a key factor in the transcription of pathological signals by increasing the expression of *STAR*, *CYP19A1* and other steroidogenesis genes. The presence of this harmful SF1 receptor in endometriotic cells is partly caused by a lack of methylation of a CpG island (cytosine–phosphate–guanine) of the promoter region of the *SF1* gene. This DNA region is normally highly methylated in stromal endometrial cells, which induces the blocking of *SF1* expression. Thus, the increased expression of *SF1* in endometriotic tissue compared to normal endometrium is mainly controlled by an epigenetic mechanism [4,24,25].

Upregulation of *ESRβ* expression is subjected to a similar epigenetic process [22]. ESRβ binds to the promoter of the *ESRα* gene and downregulates its expression, ultimately promoting the removal of ESRα. The high ratio of ESRβ/ESRα in endometriotic cells leads to an enhanced ESRβ level. ESRβ binds to the promoter of the progesterone receptor and downregulates its expression (Figure 4) [4].

Furthermore, during embryonic differentiation of the female reproductive system, environmental factors (e.g., endocrine disruptors) or genetic factors could cause genetic changes in DNA methylation. Consequently, epigenetic alterations modifying the expression of critical genes, such as *SF1* or *ESR-β* in progenitor cells destined to become various pelvic tissues, could predispose adult women to endometriosis [4,24,26]. Recently, this hypothesis was supported by Kumari et al., who highlighted a significant hypomethylation of promoter regions of proinflammatory and proangiogenic genes involved in the molecular pathophysiology of endometriosis development. This hypomethylation could be the reason for their overexpression in endometriosis [26].

In conclusion, in endometriotic cells, exposure to PGE2 leads to SF1 binding to promoters of several steroidogenesis genes (particularly aromatase) and causes the formation of large amounts of estradiol. Estradiol acts through its ESRβ receptor, whose expression is increased in cases of endometriosis, and stimulates COX-2, which leads to overproduction of PGE2. Inflammation and estrogen are linked in a positive feedback cycle inducing the overexpression of genes encoding aromatase and COX-2 and continuing the formation of aromatase and the COX-2 products—estradiol and PGE2—in endometriotic tissue. Finally, the decrease in the expression of PR induced by ESRβ is partly responsible for the resistance to progesterone and the disruption of paracrine inactivation of estradiol. Large amounts of estradiol accumulate due to its increased formation and inadequate inactivation in endometriotic tissue, promoting the proliferation of endometriotic implants [4,5,7,22,23,24,25].

Furthermore, an embryologic mechanism has been proposed to explain the onset of endometriosis. The expression of HOXA10, the homeobox gene (Hox/HOX) involved in uterine embryogenesis and embryo implantation, has been highlighted in endometriotic foci outside the Müllerian tract and could play a role in the development of endometriosis by inducing the formation of ectopic endometrial cells during embryogenesis [27].

### 2.4. Telomeres, Telomerase and Endometriosis

Permanent high estrogen levels can also promote the activity of telomerase, an important enzyme involved in aging through its crucial role in cell proliferation.

Telomeres are specialized noncoding repeated DNA sequences (5-TTAGGG-3) protecting all eukaryotic linear chromosomal ends [28]. Telomeres enable progressive shortening of chromosome extremities without inducing genetic information loss, which maintains genomic stability [29]. The progressive attrition at each replication cycle leads to a critically short telomere length, which induces proliferation arrest, senescence or apoptosis of somatic cells [30]. Moreover, telomere attrition increases in inflammatory situations [31,32,33]. Since the origin of endometriosis is unclear and accumulating evidence suggests that inflammation plays a major role in this pathology, some authors have explored telomere length and telomerase activity in the context of endometriosis.

Three studies have examined the association between leukocyte telomere length and endometriosis, with controversial conclusions. One study reported longer telomeres in leukocytes among women with endometriosis compared to those without endometriosis [34]. Another study described that shorter leukocyte telomere length was associated with a high probability of having an endometriosis history [35]. The last study reported no association between peripheral blood leukocyte telomere length and endometriosis [36].

Endometrial telomere length was significantly longer than the corresponding blood telomere length, suggesting tissue-specific regulation mediated by telomerase [37]. Some authors have described increased telomerase activity in endometrial tissue from women suffering from endometriosis versus healthy women [36,37,38,39,40], which could be explained by the enhanced expression of hTERT (the catalytic reverse transcriptase subunit of telomerase) caused by the binding of estradiol to its promotor estrogen response element [41,42]. Similar to what happens in tumors, telomerase activity may promote the cellular proliferation of endometrial tissue in endometriosis. Nevertheless, the association between telomerase activity and the endometriosis stage is not clear [36,39]. Further studies are warranted to elucidate the interrelationship between telomere length and the inflammatory and hormonal background among patients with endometriosis. Direct telomerase inhibition in endometrial tissue from women with endometriosis may arrest the proliferation and dissemination of endometriotic lesions. Some authors have confirmed this hypothesis by stopping the in vitro proliferation of endometriotic cells with imetelstat, an experimental anticancer telomerase inhibitor [37].

### 2.5. Immune Escape

Several studies are in favor of the phenomenon of cellular immune escape allowing endometriotic cells to proliferate. Some arguments are inherent to ectopic endometrial cells, while others depend on the immune system.

First, grouping endometriotic cells into fragments protects cells located in the deeper layers of these fragments. Moreover, endometriotic cells have different characteristics that can allow them to escape this immune system: (i) they expose modified type I HLA antigens; (ii) they secrete TGF-β and PGE2, responsible for lymphocyte inhibition; and (iii) they secrete HLA soluble antigens or sICAM-1. These factors confer a greater resistance to lysis from NK cells because they bind to NK’s LFA-1 receptor (competition with ICAM-1 membrane receptor). Finally, they can induce apoptosis of immune cells via mechanisms involving the Fas system [2,5,7].

Furthermore, the immune system of patients with endometriosis is suspected to be dysfunctional. In particular, NK cells have altered activity [7]. Another argument in favor of immune dysfunction in the development of endometriosis is the high prevalence of associated autoimmune diseases (SLE, rheumatoid arthritis, SGS, and autoimmune thyroid diseases) and atopic (allergies, eczema and asthma) diseases in these patients [5,7].

### 2.6. Adhesion, Implanting and Invasion

Endometriosis appears and develops according to the following stages: (i) reflux; (ii) adhesion; (iii) proteolysis; (iv) proliferation; (v) angiogenesis; and (vi) lesion [2].

A constitutional or acquired altered peritoneum would be a predisposing factor to attachment and to mesothelium invasion since an intact mesothelium is a natural barrier to this pathological process. In vitro studies have shown that fragments are only implanted at peritoneal locations where the extracellular matrix and basal membrane are exposed due to damage in the mesothelial layer. Therefore, retrograde menstruation would have a harmful effect, thus explaining the occurrence of such mesothelial damage favoring implantation [2,5].

At the molecular level, first, a strong interaction between hyaluronic acid of the extracellular matrix and CD44 of endometrial cells initiates the adhesion process [7]. Moreover, some fibronectin receptors (α4β1, α5β1), whose endometrial expression normally varies according to cycle phase and estrogen levels, are constantly expressed by endometriotic cells. This suggests a potential role of these receptors in cell adhesion [2].

Second, implantation is promoted by the inflammatory environment resulting from the overexpression of matrix metalloproteinases (MMP-1, MMP-2, and MMP-3) and ICAM-1 during the luteal phase, as well as the increased levels of TGFβ, IL-6, IL-1β and TNFα. MMPs and their inhibitors (TIMPs) are involved in extracellular matrix remodeling. Their expression varies with the phase of the cycle and suggests ovarian hormonal regulation. Most isoforms of MMPs are synthesized and activated during the endometrial proliferation phase, particularly under stimulation by estrogen (in contrast, progesterone tends to decrease their synthesis). The balance between MMPs and TIMPs is essential for ensuring the correct MMP activity. MMP hyperactivity could lead to matrix disruption and thus cell invasion. In women with endometriosis, the TIMP-1 concentration is precisely decreased in peritoneal fluid. Moreover, the expression and activity of MMP-7, MMP-1 and MMP-3, normally reduced by progesterone during the ovulatory phase, persist in endometriotic lesions due to progesterone resistance. Furthermore, deregulation of the E-cadherin system of endometriotic cells makes it possible to initiate the invasion process, similar to what is observed in carcinoma cells [2,4,5,7].

### 2.7. Growth and Lesional Neuroangiogenesis

Oxygen and various nutrient supplies that are essential for the growth of endometriosis implants are transported due to angiogenesis’ vascular development. This neovascularization in a physiologically avascular peritoneum induces a rather favorable environment. The concomitant development of nerve fibers explains the pain experienced by patients [3,43]. The initiation of the phenomenon involves the secretion of several cytokines (especially by peritoneum macrophages), namely TNFα, TGF-α, TGF-β, IL-8, MMP-3, and mainly VEGF (its peritoneal fluid rate is correlated with disease severity). VEGF is mainly synthesized during the secretory phase of the menstrual cycle in a healthy endometrium. In the case of endometriosis, elevated levels of VEGF have been reported in the peritoneal fluid during the proliferative phase of the menstrual cycle (when the peritoneum is exposed to retrograde menstruation). Furthermore, factors modulating its secretion (localized hypoxia, IL-1β, TGF-β, EGF and PGE2) are increased in the case of endometriosis. In addition to promoting angiogenesis, they also increase capillary permeability, facilitating macrophage diapedesis. Other mitogenic factors for endometriosis cells are involved, such as angiogenin, platelet-derived endothelial growth factor (PEGF), macrophage migration inhibitory factor (MMIF), hepatocyte growth factor (HGF), epidermal growth factor (EGF), insulin-like growth factor (IGF) and basic fibroblast growth factor (bFGF) [2,5,7,14].

### 2.8. Inflammation

In the case of endometriosis, the peritoneum shows an increased number of activated macrophages (with increased activity) and high levels of many cytokines, such as MMIF, TNFα, IL-1β, IL-6 (largest proportion), and IL-8. However, it is difficult to conclude whether these phenomena are a cause or a consequence of endometriosis [5,7,14].

Chemokines, such as monocyte chemoattractant protein 1, IL-8 and regulated upon activation normal T-cell expressed and secreted (RANTES), are involved in chemotaxis, promoting the influx of polynuclear neutrophils, NK cells, and peritoneal macrophages. Positive autoregulation (positive feedback) maintains this phenomenon and causes both an accumulation of immune cells and elevated levels of cytokines in endometriotic lesions [2,4,7].

This positive feedback is further accentuated by the hormonal climate of endometriosis in the following way: In women suffering from endometriosis, peritoneal fluid macrophages showed a significantly greater ability to secreteCOX-2 and therefore to secretePGE2. Furthermore, TNFα promotes the production of PGF2a and PGE2 by endometriotic cells, while IL-1β activates COX2, inducing PGE2 production and consequently activating aromatase. E2, resulting from high aromatase activity (which is also increased by MMIF, contributing to positive feedback), induces the increased synthesis of IL-6 and TNFα, leading to maintenance of the proinflammatory context [14]. On the other hand, anti-inflammatory progesterone action is lacking in cases of endometriosis because of progesterone resistance. Inflammation in women with endometriosis is not only limited to endometriotic peritoneal lesions but is also found throughout the endometrium [5,14].

Clinical studies revealed that endometriotic stromal cells release cytokines (IL-33 and others) and promote a type 2 immune response [44]: macrophages transform into M2 subtype, and T regulatory cells (Tregs) into Th2-like Tregs which secreted high levels of IL-4, IL-13, TGF β1.

In association with the platelet-BB-derived growth factor of platelets, these cytokines directly and indirectly, via endometriotic cell-Tregs interference, promote the emblematic fibrogenesis of endometriosis [45].

Interleukin-17 (IL-17), secreted by CD4+ T helper 17 (Th17cells), is another proinflammatory cytokine involved in the regulation of the immune microenvironment of endometriotic lesions: IL-17 promotes proliferation, invasion, and implantation of endometriotic cells directly and indirectly through the recruitment and activation of neutrophils (via IL-8 and granulocyte-colony stimulating factor (CSF) and granulocyte macrophage-CSF). IL-17 recruits and activates M2 macrophages, which, in response, release nitric oxide. In addition, IL-17 recruits lymphocytes and bone-marrow-derived cells, inducing the secretion of proangiogenic factors [46].

### 2.9. Role of miRNAs in Endometriosis

MicroRNAs are small noncoding RNAs modulating gene expression through mRNA degradation or other interactions. Involved in almost all diseases, they have also been investigated in the pathophysiology of endometriosis [2,4,5,7,47]. Some miRNAs regulate the epithelial-mesenchymal transition [48], essential for the dissemination of epithelial cells. Others would modulate the hormonal environment (miRNAs -23a and -23b, miRNAs -:135a, 135b, 29c, and 194 −3p) via their interaction with SF1. Others promote angiogenesis (miRNA -126, miRNA -210, miRNA -21, miRNA -199a-5p and miRNAs 20A). Others increase inflammation and cell proliferation (miRNA -199a and miRNA -16). Others are the consequences of modified environmental conditions such as oxidative stress (miRNA -302a). Some miRNAs have been proposed as diagnostic biomarkers since their concentration in blood is increased significantly in case of endometriosis [47,49].

## 3. Endometriosis and Cancer

The hyperestradiolaemia associated with endometriosis suggests a potential increase in female genital cancer.

A recent meta-analysis concluded a positive association between endometriosis and ovarian cancer [46]. Endometriosis doubles the risk of developing ovarian cancer (SRR = 1.93, 95% CI = 1.68–2.22; *n* = 24 studies), maximizes the occurrence of clear cell histotype of ovarian cancer (SRR = 3.44, 95% CI = 2.82–4.42; *n* = 5 studies), and moderately increases that of endometrioid histotype (SRR = 2.33, 95% CI = 1.82–2.98; *n* = 5 studies). The type of endometriosis is also a crucial element: endometrioma multiplies by 5.41 the risk of ovarian cancer. Since cancer-driver mutations (*KRAS*, *ARID1A*, *PIK3CA*) are similar in deep lesions and endometriomas, tumorigenesis results from additional factors. Ovarian stromal conditions are particularly well-suited for the proliferation of endometriotic cell clones [17]. Estradiol levels are very high in ovarian stroma. In endometriotic stroma cells, the very active CYP1B1 converts estradiol to 4-OH-estradiol, which is further converted to 4-OH-estradiol-quinone damaging DNA via alkylation or oxidation, promoting mutations in addition to the cancer driver mutations described earlier.

Interestingly, endometriosis was associated with a minimally increased risk of breast cancer (less than 10%). Only rare studies reported an increased risk for estrogen receptor-positive (ERþ)/progesterone receptor-negative (PR) breast cancer (ERþ/PR: HR = 1.90, 95% CI = 1.44–2.50) [50,51].

The association with endometrial cancer is controversial [50,51] probably because some biases complicate analyses. For instance, patients with lean size (low BMI) have an increased risk of endometriosis [15]. In contrast, high BMI increases the risk of endometrial cancer probably because of the high production of testosterone by fat tissue and because of abnormal insulin pathways [52].

The unpredicted inverse correlation of endometriosis with cervical cancer (SRR= 0.68, 95% CI =0.56–0.82; *n* = 4 studies) potentially results from better access to early diagnosis and treatment of cervical lesions in patients with endometriosis because the painful character of the disease leads patients to consult their gynecologist more frequently. Moreover, chronic pelvic pain and dyspareunia may limit the sexual relationships of patients with endometriosis and, therefore, their contamination with HPV [50].

Endometriosis also increases the risk of thyroid cancer (SRR = 1.39, 95% CI =1.24–1.57; *n* = 5 studies) but not of colorectal cancer (SRR = 1.00, 95% CI =0.87–1.16; *n* = 5 studies).

The association with cutaneous melanoma was controversial [50].

## 4. Endometriosis and PolyCystic Ovaries Syndrome (PCOS)

Polycystic ovaries syndrome is characterized by multiple cysts at the surface of the ovaries in association with endocrine and metabolic disorders. The endocrine syndrome results from the high production of estrogens and androgens and insulin resistance with overweight and diabetes mellitus.

PCOS and endometriosis share a common association with high ovarian estrogen levels. In both cases, a hormonal balance of sex hormones is disturbed: high estrogen with progesterone resistance in endometriosis, and high estrogen with high androgen in PCOS.

Experimental animal studies and human epidemiologic studies support a developmental theory of both diseases. Both would result from an abnormal fetal androgen impregnation during the in utero programming of the hypothalamic–pituitary–gonadal (HPG) axis with other environmental and genetic factors. Low prenatal testosterone results in the programming of the female fetal HPG axis, leading to features associated with endometriosis profile, such as early puberty, low LH/FSH rate, low AMH, fast folliculogenesis, and short anogenital distance. In contrast, high prenatal testosterone orientates the female fetal HPG axis in the opposite direction: late puberty, high LH/FSH rate, long folliculogenesis, and long anogenital distance. This hypothesis is supported by the relatively rare prevalence of diseases together [53].

Multiple PCOS follicles and cysts produce high levels of estrogen. Chronic anovulation favors ovarian accumulation, which may increase the risk of ovarian cancer. This hypothesis is supported by similar DNA hypomethylation and miRNAs in PCOS ovaries and ovarian cancer. In addition to high estrogen levels, PCOS ovaries secrete high testosterone levels, increasing the risk of endometrial cancer [54,55].

## 5. Infertility and Endometriosis

Various mechanisms may explain the consequences of endometriosis on several steps of reproduction, especially: (1) the tubal transfer of the oocyte-cumulus complexes; (2) gamete interaction; (3) implantation; (4) the importance of ovarian reserve and oocyte quality; and (5) sexual behavior.

First, the distorted pelvic anatomy in relation to major pelvic adhesions can disrupt oocyte release from the ovary and disturb the tubo-uterine passage [4,9].

Second, peritoneal fluid is more abundant, and its modified composition is the consequence of endometriotic lesions, as mentioned before (see the section on inflammation). This liquid is largely in contact with the bulb at the distal end of the fallopian tubes close to the fertilization site. Therefore, its chemical composition can directly influence and disturb gamete interactions. In particular, IL-1 and IL-6 directly affect sperm motility. TNFα induces DNA damage through reactive oxygen species (ROS) (resulting frequently in cell apoptosis). These cytokines could also prevent sperm capacitation. Finally, the oxidative stress induced by ROS inhibits the acrosome reaction and gamete fusion [9,56].

As mentioned earlier, M2 macrophages activated by peritoneal IL-17 release a high amount of nitric oxide (NO) [46] with an additional harmful effect on sperm, embryo development and implantation. Reducing NO synthesis in peritoneal fluid or blocking the effects of NO could limit the impact of endometriosis on fertility [57].

Third, in the endometrium, the influx of immune cells results in an increased release of several cytokines, modifying the endometrial environment as explained above [56]. Lymphocytes secreting IgG and IgA autoantibodies can disturb embryo implantation [9,58]. Other studies have shown reduced expression of αvβ3 integrin, ensuring physiological cell adhesion [9].

Fourth, in the ovaries, inflammatory endometriotic cysts can damage the ovarian cortex and decrease follicular reserve. This phenomenon can be accentuated by the surgery performed in this case [9,56]. In women with endometriosis, lower oocyte and embryo quality is frequently observed. Embryos derived from oocytes from women with endometriosis show decreased implantation rates, even when the transfer is carried out in a uterus without endometriosis (healthy women). However, these findings should be confirmed in further studies [9].

Finally, chronic pelvic pain induced by pelvic inflammation and adhesions causes dyspareunia, leading to a reduced frequency of sexual intercourse. This behavioral phenomenon significantly reduces the chances of natural conception [59].

## 6. Environmental Impact

The fetal environmental impact on the subsequent genesis of various pathological processes at an early stage of development has long been studied as the Barker hypothesis [60]. As detailed below, many authors, including Bulun et al., defend the hypothesis of an epigenetic process at the origin of the pathophysiology of endometriosis [4,61].

Prenatal exposure to multiple ubiquitous pollutants or toxic molecules is well-established [62]. It obviously includes cigarette smoke but also chemicals belonging to the category of endocrine disruptors. In this category, there are compounds with short half-lives, such as bisphenol A (BPA) or phthalates, and compounds with long half-lives, such as dioxins. Finally, some hormones and some drugs with hormonal action were suspected of promoting the endometriosis process, especially diethylstilbestrol (DES) and ethinyl estradiol (EE) [61].

### 6.1. EE and DES

By orally administrating high amounts of EE to mice from the 11th to the 17th days of gestation, Koike et al. showed that this experimental prenatal exposure increased the incidence of endometriotic lesions in the next generation [63].

DES, prescribed to millions of women between 1938 (discovery date) and the 1970s (when its use was banned) to limit or prevent recurrent miscarriages, was responsible for a large number of deleterious effects in the in-utero-exposed fetus [62]. DES had a very strong and long EE-like activity. In the 1980s, Haney and Hammond studied the influence of DES on fertility. In a small group of 33 infertile couples with women exposed to DES in utero, they found that infertility was due to the presence of endometriosis in 11 of them. The reported association is not clearly established because the study did not include an infertile population control group [64]. More recently, in a prospective cohort study, Missmer et al. (“Nurses Health Study II”) identified a higher relative risk (RR = 1.8, CI = 1.2–2.8) of developing endometriosis in women exposed to DES in utero [10]. For other authors, such as Benagiano and Brosens, the impact of endometriosis would be greater in women exposed to DES [65]. However, Wolff et al. failed to confirm a significant association between DES and endometriosis in the “ENDO study”. This large study involved a cohort of 473 patients operated on through laparoscopy and a control cohort of 127 patients who completed a pelvic magnetic resonance imaging, all from 40 clinical investigation centers in Utah and California between 2007 and 2009 [66].

The exact mechanism of DES on endometriosis development is unknown. Some authors proposed a link between in utero DES exposure and cervical stenosis, smooth uterine muscle abnormalities or altered expression of estrogen receptors [67,68,69,70]. Golden et al. explained that high exposure to estrogen (or derivatives such as DES) during embryonic development could cause a disruption of genes under the influence of steroid hormones (such as genes encoding ESR) [71]. Koike et al. showed that in mice exposed to DES, constant expression of the lactoferrin and *EGF* genes can be observed in the vagina and uterus [63]. Furthermore, Wang et al. hypothesized that EGF could stimulate endometriosis cell proliferation by activating the Ras/Raf/MEK/ERK pathway [72,73,74].

### 6.2. Dioxins

Dioxins are chlorinated and polycyclic aromatic lipophilic agents and can persist for a long time in organisms. This results in a bioaccumulation process. These compounds, which are only produced by human activities, include dioxins and “dioxin-like” compounds: polychlorinated dibenzo-p-dioxins (PCDDs) or dioxins, polychlorinated dibenzofurans (PCDFs) and polychlorinated biphenyls (PCBs) [1,75,76,77,78]. 2,3,7,8-p-Dioxin-tetrachlorodienzo (TCDD) is the most toxic dioxin, and its toxicity is also a reference for assessing the impact of other organochlorine compounds with the toxic equivalency factor (TEF) [79].

TCDD can alter specific ESR rates but also, more generally, the metabolism of steroid hormones. Its toxic effects are directly related to its binding to the nuclear receptor AhR (aryl hydrocarbon receptor), leading to the formation of an activated heterodimer with transcriptional action. Finally, overexpression of AhR may result in an inflammatory state and promote menstruation. The activation of AhR may also activate other factors involved in cell proliferation, such as TGFβ [1,76]. Moreover, TCDD increases the secretion of MMPs and can also induce progesterone resistance [1,75]. Otherwise, TCDD might act by disrupting the expression of microRNAs [80]. Studying TCDD exposure in mice, Bruner-Tran et al. concluded that in utero TCDD leads to a phenotype of progesterone resistance persisting over several generations [81].

According to Cummings et al. [82], PCB exposure is associated with a higher risk of developing endometriosis via a mechanism similar to that previously described with TCDD via a pathway involving AhR. PCB also decreases circulating NK cell activity, as well as the production of IL-1β and IL-12 [75].

The first epidemiological and clinical studies on the impact of dioxins and endometriosis development were conducted by Rier et al. [83]. The same authors have shown a link between TCDD and endometriosis in rhesus monkeys.

Thereafter, other studies have been conducted, in particular to check the consequences of the Seveso catastrophe in 1976: Eskenazi et al. have not demonstrated an increased incidence of endometriosis in patients living in the Seveso area after the disaster [84]. Globally, the literature is controversial: some authors have shown a potential link between endometriosis and dioxin exposure [82,85,86,87,88,89,90,91], and others have shown no correlation [92,93,94,95,96,97,98,99,100,101]. There is no publication on the potential involvement of Agent Orange in the onset of endometriosis in Vietnamese women after the Vietnam War. Finally, some studies have suggested an inverse relationship, that is, a “protective” role of dioxins. Furthermore, a study highlighted a lower incidence of endometriosis in children breastfed with possible exposure to dioxins in maternal milk than in adults [102].

Because the epidemiological and analytical methodologies are so different, these conflicting findings do not suggest a potential link. However, recent studies support the possible role of dioxin epigenetic modification in endometriosis [103,104].

### 6.3. Bisphenol A

BPA is used in industry as a monomer for epoxy resins (tins, cans) and polycarbonate (plastic industry, additives), and has estrogenic activity.

Upson et al. used data from the “Women’s Risk of Endometriosis study” and a control population, including 143 endometriosis patients and 287 controls, to show that BPA exposure is associated with a higher risk of developing endometriosis [105]. Signorile et al. studied the effects of BPA in a mouse model. According to their study, animals exposed to BPA present a higher incidence of adenomatous hyperplasia with cystic endometrial hyperplasia, atypical hyperplasia and ovarian cysts (45–50%) than control animals (10%) [106].

BPA action leads to a hyperestrogenic environment via inhibition of the expression of PR and progesterone activity and the promotion of E2 activity. Such hormonal alterations during a critical period of embryogenesis may increase the susceptibility to developing endometriosis and generally other diseases through an epigenetic mechanism [75,106].

According to Xue et al., BPA promotes endometriosis by facilitating endometrial stromal cell invasion [107], especially by upregulating matrix metalloproteinases 2 and 9. Moreover, Xue et al. highlighted the upregulation of Erβ expression in endometrial cells via the WD repeat domain 5/TET methyl-cytosine dioxygenase 2 (WDR5/TET2)-mediated epigenetic pathway [108].

Environmental BPAF, a fluorinated homolog of BPA with stronger estrogenic activity, may promote, alone or in association with BPA, the development of endometriosis [109].

### 6.4. Phthalates

Chemicals derived from phthalic acid, namely phthalates, are commonly used in the plastics industry. With approximately 3 million tons produced worldwide per year, phthalates and phthalate metabolites are present everywhere at different rates in our environment: cosmetics, paints, clothes, toys, etc. Several phthalates have been classified as toxic substances for human reproduction (CMR category 1B) by the European Chemicals Agency (ECHA) [110]. In the atmosphere, their physicochemical properties ensure easy transport and thereafter potential bioaccumulation in the food chain (especially for low-molecular-weight phthalates) [111].

Their (repro)toxicity is manifested through toxic effects on sperm, early puberty in girls, abnormalities of the genital tract, and infertility, in addition to adverse effects on neurodevelopment or simply allergies [111].

Several studies have focused on the mechanism of the toxic action of phthalates in the development of fish embryos, especially di(2-ethylhexyl) phthalate (DEHP), diethyl phthalate (DEP), dibutyl phthalate (DBP) and benzyl butyl phthalate (BBP). For these molecules, oxidative stress would be the most critical mechanism of toxicity (CMTA = Critical Mechanisms of Toxic Action) in the case of DEHP and DEP exposure. Unlike DES or dioxins, phthalates do not act as E2 receptor agonists and have a very low affinity for AhR [111].

In a recent review, Kim and Kim summarized the mechanisms through which phthalates, especially DEHP, promote endometriosis: (1) phthalates induce a modification of estrogen receptor type; (2) they increase the resistance to apoptosis of endometriotic stroma cells; (3) they increase the invasiveness of endometriotic stroma cells through the stimulation of MMP2 and 9 secretion; and (4) they cause oxidative stress and reduce antioxidant enzymes, finally leading to an enrichment of environmental ROS. All these mechanisms increase the proliferation and invasiveness of endometriotic stromal cells, promoting endometriosis [112].

Several studies have explored the relationship between endometriosis and phthalate exposure; however, they were biased due to contamination with phthalates from the collection tubes and other equipment and laboratory supplies. Three epidemiological studies conducted by Huang et al., Itoh et al., and Weuve et al. assessed the risk of endometriosis in relation to the urinary concentration of phthalate metabolites [113,114,115]. Unfortunately, the results are conflicting. Upson et al. resumed the principle of study and obtained conflicting results: an inverse correlation between endometriosis and urinary phthalate levels, while they observed an increased risk of endometriosis with high urinary levels of mono-benzyl phthalate (MBzP) and mono-ethyl phthalate (MEP); however, the results were not statistically significant [116].

In the previously mentioned “ENDO study”, Buck Louis et al. did not report a strong correlation between endometriosis and phthalates. However, the short half-life of phthalates in the blood can lead to an analytical bias with this type of assay (urine being preferred) [117].

A recent meta-analysis concluded that there was a potential statistical association only between MEHHP exposure and endometriosis, particularly in Asia, but not between other phthalate acid esters (PAEs) and endometriosis. The authors acknowledged the weak strength of the results due to the lack of well-designed cohort studies with large sample sizes [112].

### 6.5. Tobacco

The effects of tobacco on the development of endometriosis are controversial. Smoking seems to be a protective factor against the endometriosis process. Several studies have demonstrated an inverse correlation between tobacco consumption and endometriosis [118,119]. However, Haney and Hammond [64] and Somigliana et al. [92] failed to demonstrate this association. The large cohort study of Hemmert [120] confirmed the meta-analysis of Bravi et al., which concluded the absence of a link between tobacco consumption and endometriosis [121].

The protective effect of tobacco has been suggested for many years via the hypoestrogenic action of some tobacco compounds, namely nicotine and cotinine (one of the major metabolites of nicotine), which influence the metabolism of steroid hormones and prevent the conversion of androgens to estrogens. Furthermore, nicotine promotes cell apoptosis, with the consequence of limiting the proliferation of endometriotic cells. Finally, this substance induces a decrease in the cell inflow and activity of NK cells to limit the inflammatory phenomenon that is well-described in endometriosis [75,118]. Experimental animal data and human ex vivo experimentation suggest a role of C-X-C motif chemokine ligand 12 (CXCL12) and fibroblast growth factor 2 (FGF2), two cytokines with a pro-cell proliferative action, whose secretion is decreased in the case of tobacco consumption [122].

Moreover, increasing the secretion of VEGF by nicotine is suspected to support the development of endometriosis by promoting neoangiogenesis and vascularization of endometriotic lesions [75].

Furthermore, secondhand smoke during childhood due to maternal smoking seems to be associated with an increased risk of endometriosis in adolescents and young adults [123].

## 7. Role of the Microbiota in Endometriosis

Recently, the study of the microbiota to decipher the physiopathology of many complex pathologies has been applied to endometriosis. The literature has rapidly expanded in the last eight years. Some reviews have tried to summarize how the microbiota regulates factors involved in maintaining the normal peritoneal environment and ectopic cell clearance, and how dysbiosis contributes to the dysregulation of factors driving endometriosis development [124]. A specific composition of the gut microbiota is suspected to induce immune dysregulation, which can progress into a chronic state of inflammation, a perfect environment for endometriosis progression. Endometriotic microbiotas have been consistently associated with diminished Lactobacillus dominance on the one hand, and an altered Firmicutes/Bacteroidetes ratio associated with a high abundance of vaginosis-related bacteria on the other [124,125]. In comparison, in PCOS, the distortion of microbiota results in an abnormal Escherichia/Shigella ratio and an excess of Bacteroides [126].

Some studies even suggest a main infectious origin in the pathophysiology of endometriosis [127,128,129].

Furthermore, estrogen metabolism is known to be regulated by the estrobolome, a collection of gut bacteria involved in estrogen metabolism. Estrobolome activity modulates the amount of excess estrogen that is excreted from or reabsorbed into the body. When this activity is impaired, especially in cases of imbalances in the gut microbiome, excess estrogen can be retained in the body and diffuse from the gut to the endometrial and peritoneal environment via the circulation. This contributes to the hyperestrogenic environment that drives endometriosis and provides a possible mechanism as to how dysbiosis in the gut microbiota may be involved in the disease [130,131].

Interestingly, women with a high intake of omega-3 polyunsaturated fatty acids (PUFAs) have a lower risk of endometriosis. A similar diet showed anti-inflammatory effects and suppressed the formation of endometriotic lesions in murine models. This suggests an at least partial contribution of diet to the induced modification of the gut flora [124,132].

This concept of the relationship between the microbiota and endometriosis leads to the consideration of antibiotics as a new promising approach for endometriosis treatment. In animal models, broad-spectrum antibiotics have already proven efficacious for treating endometriosis. In a recent murine study, broad-spectrum antibiotics inhibited ectopic lesions, while treatment with metronidazole significantly decreased inflammation and reduced lesion size, possibly by lessening Bacteroidetes presence [133]. Alternatively, probiotic intervention, that is, the administration of live microorganisms, could be another effective approach [134,135].

Since most chemical endocrine disruptors transit the digestive tract, they interact with gut microbiota. On the one hand, endocrine disruptors can modify the microbiota or modulate microbiota enzymatic activity. In the long term, an endocrine disruptor can alter the microbial diversity of the microbiota. On the other hand, microbiota metabolizes part of chemicals, therefore modulating their toxicity [136]. Prenatal exposure to endocrine disruptors may promote endometriosis via altered maternal and fetal microbiota [137], resulting in abnormal sex hormone levels (as exposed earlier). An alteration of microbiota may also decrease the potential of DNA methylation since some gut bacteria produce folate, a central methyl donor [138].

## 8. Conclusions

Endometriosis is a gynecological disease with a complex pathophysiology (Figure 5). To date, the specific pathogenesis of endometriosis has not been clarified, and some recent studies have suggested a potential role of the gut microbiota [139]. What is certain is that there is a key role of estradiol and retrograde menstruation. Endometrial tissue transformation in endometriosis can be observed in women exposed in utero to endocrine disruptors. These substances are the root cause of an epigenetic process disrupting the expression of key steroidogenesis genes in endometrial cells.

Epidemiological studies of exposure to the molecules probably involved in such a mechanism, such as dioxins, bisphenol A, phthalates, DES, or nicotine, have not found strong and repeatable correlations. However, these studies are conducted using heterogeneous methodologies. Measurement errors in estimating the behaviors during pregnancy of mothers of daughters suffering from endometriosis can hinder the results obtained because of “subjectivity” and recall bias. All the abovementioned studies are not based on the same clinical diagnosis repositories of endometriosis because they were not performed at the same time. Selection bias could have occurred. In addition to classification errors, all of these aforementioned elements constitute an important limitation [10,66].

Further large-scale and homogeneous studies are needed to draw conclusions about the influence of these endocrine-disrupting compounds on the development of endometriosis.

## Figures and Tables

**Figure 2 biomedicines-11-00978-f002:**
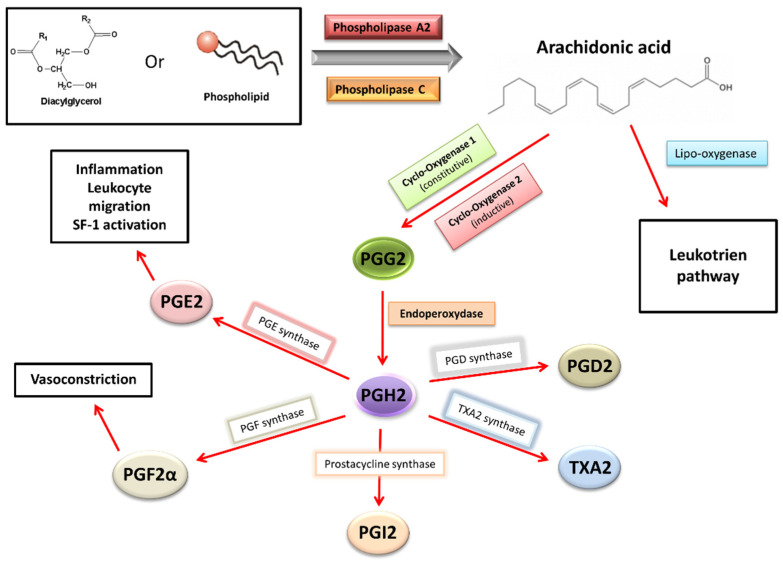
Prostaglandins synthesis and effects in endometriosis [4,5].

**Figure 3 biomedicines-11-00978-f003:**
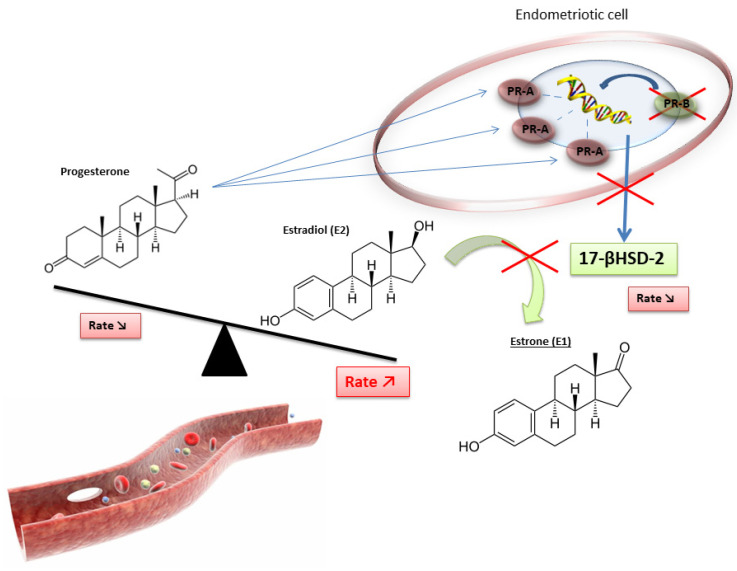
Disruption of progesterone/estradiol balance in endometriosis [4].

**Figure 4 biomedicines-11-00978-f004:**
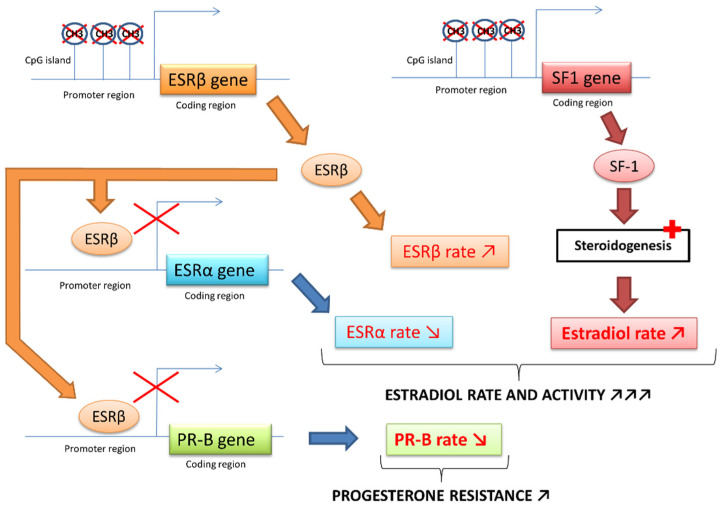
ESR and SF-1 receptors epigenetic modifications in endometriosis [4,22,23].

**Figure 5 biomedicines-11-00978-f005:**
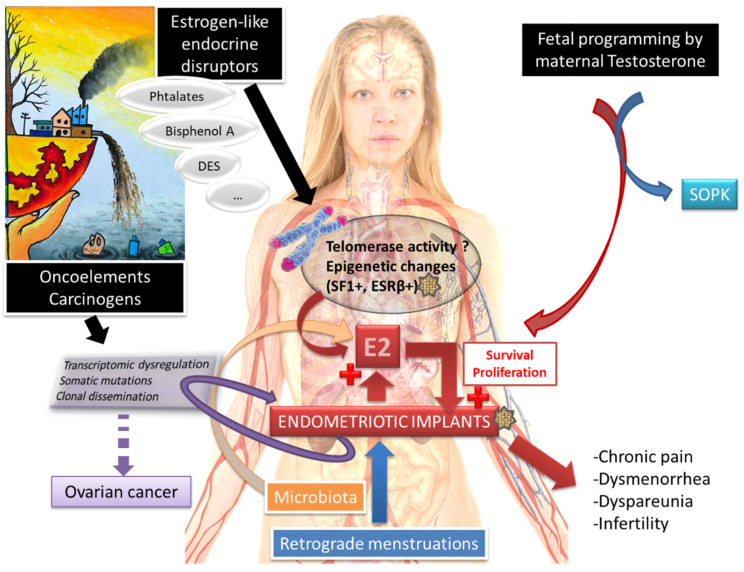
Endometriosis, a complex disease, with several concomitant etiologies?

## Data Availability

All references cited in the manuscript are available in pubmed.

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
