# Peer review of "Endometriosis: Update of Pathophysiology, (Epi) Genetic and Environmental Involvement"

_biomedicines, 2023, doi:10.3390/biomedicines11030978_

Round 1

Reviewer 1 Report

Endometriosis is a common gynecological disease and it is also a complex ailment with a multifactorial pathogenesis. The authors summarized the
p
athogenesis and pathophysiology of endometriosis.

My minor concerns:
1.
There are mountain studies conducted to discuss pathogenesis and pathophysiology of endometriosis. What is already known on this topic, and what does your manuscript add?

2. Line 81: The role of cancer-associated mutations in the development of
endometriosis
should be discussed in the text. Recent published article on this
topic
(Nat Genet. 2023;55(2):255-267; Fertil Steril. 2022;118(3):524-534) should discussed in the text.

3. The diagnostic value or the role of microRNAs in endometriosis can also be added.

4. Line 345: Recent published articles such as Biomedicines. 2022;10:2893 and Int J Med Sci. 2022;19:769-778 should be added and discussed in the text.

5. The tile is Endometriosis, Environment and Genetics” but the content of the
review article
contains too many risk factors for developing endometriosis. I
suggest the authors
rewrite the title or reorganize the text to fit your title.

Reviewer 2 Report

The paper is very interesting with high clinical impact. Endometriotic implants induce inflammation, leading to chronic pain and impaired fertility.  The data concerning PCOS and endometrosis should be added and discussed  since microbiota  is probably crucial  for women with these diseases. Oncoelements might be also important in the context of endometriosis and enviroment. 

Round 2

Reviewer 1 Report

The authors have addressed all my comments. I have no further questions.